# Latin American Megachurches in a Changing Culture: An Integrative Review and an Exploration of Future Research Directions

Fernando Adolfo Mora-Ciangherotti

Bioengineering and Applied Biophysics Group, Universidad Simón Bolívar, Caracas 89000, Venezuela; fmora@sgu.edu

**Abstract:** Several review articles about megachurch studies have been published recently concentrating their work on USA, Europe, and other parts of the world, with just a few references about Latin American megachurches. For that reason, this article aims to identify some of the characteristics of Latin American Evangelical megachurches by looking at relevant literature, especially that produced in the region, in Spanish and Portuguese, which is usually overlooked by researchers in the Global North. Since this research field is still limited in Latin America, areas where further work is necessary are identified. Three general catalysts for the emergence of megachurches in the region, *church growth methodologies*, *Pentecostalization*, and *theologies of growth*, serve as guides to organize the review process. The discussion shows that several potential areas of research in a variety of fields such as theology, ecclesiology, organizational theory, leadership, gender studies, and ethics, are proposed from the review.

**Keywords:** megachurches; church growth; Pentecostalization; prosperity theology; dominion theology; cell groups; contemporary worship music; spiritual warfare

## 1. Introduction

The growth of megachurches in Latin America is an interesting case of how a marginal, underestimated religious group, represented by a myriad of small congregations in the periphery of megacities and in rural areas, progressively becomes a protagonist of political and social processes in the different countries of the region. These congregations were formed outside the locus of control of North American and European missions and denominations, many of them being indigenous and independent, connected through complex church networks. These aspects and the sociological and anthropological characteristics of the Latin American context make the field of megachurch studies in Latin America unique. However, in the current literature about megachurches this trajectory is overshadowed by the dominant narratives about megachurches in the Global North as the main sources of innovation and of theological reference. In Latin America, those growing congregations, with Sunday attendance beyond the 2000-person landmark, with their culture, methods, and sociopolitical influences, "have become the most attractive churches within the contemporary evangelical landscape, especially for younger emerging social classes" (Ihrke-Buchroth 2013). This article aims to identify some of the characteristics of these Evangelical megachurches by looking at the literature published, especially from Spanish- and Portuguese-speaking countries, which is usually overlooked by researchers in the Global North.

Recently, the field of megachurch research has had several reviews and collective works that are contributing to its consolidation around the world (Bauman 2022; Hunt 2020; Cartledge et al. 2019). Although these publications stress that the number of Latin American megachurches is increasing in correlation with the impressive growth of Protestant and Pentecostal Evangelicalism in the region, the research efforts to understand the nuances

of such expansion remain frustratingly low. It is important to observe the evolution of Evangelical and Pentecostal congregations, from the margins of society to the adoption of the megachurch model, where they end up becoming new important religion centers with societal implications that are still not very well-understood. In the case of Latin American Evangelical and Pentecostal megachurches, we can see the three dimensions proposed by Robbins (2004) of *world breaking* cultural forms that guide everyday life, *world making* novel organizational structures that are created by them in order to multiply and expand, and the effect of *globalization* on the political, economic, and social agendas of these congregations guided by evolving theologies. Although these dimensions were developed to explain the process of Pentecostalization, in the case of Latin America, their observation through time provide a window to understand the relationship between religion and modernity in a traditionally Christian continent that has not undergone a profound secularization yet. The increasing presence of large Pentecostal and Evangelical churches in the forefront of social and political issues of Latin American society makes it necessary to describe how these three dimensions are expressed in the megachurches throughout the region, as well as to better understand the field and its different ramifications. Nevertheless, these topics have remained marginalized from research in the region, although there has been a slow growth of interest in recent years due to the large percentage of the Latin American population that has embraced Pentecostalism as their religion. A lack of established research groups, minimal research funding for these topics, underestimation of religious studies in universities around the region, and the overlook of Spanish/Portuguese-language literature originating in Europe and the USA influence the few articles cited in the reviews mentioned. As Paul Freston (2007) has expressed, it is important to talk more about Christianity outside the Global North, as its growth centers are "now located mainly in the Global South (Latin America, Africa, Asia, the Pacific), distant from power and wealth, and it spreads largely as an autonomous social movement and as a 'globalization from below'". In this regard, this article is an attempt to make visible some of the research efforts performed so far and to propose new avenues for investigation that show how megachurches are contributing to the transformation of Latin American Pentecostalism from the margins to become a globalized force.

## 2. From the *Barrio* Church to the Megachurch

The prevailing model of Evangelical and Pentecostal churches in Latin America was a network of small garage congregations, or groups meeting in refurbished low-income houses, or rented small commercial spaces in poor neighborhoods of growing cities, which were sprouting everywhere, led by amateur pastors with limited hands-on biblical training. Historically speaking, the first Protestant congregations in Latin America were the product of the missionary zeal of traditional Evangelicalism (Gladwin 2019), which planted many churches and started seminaries and other educational initiatives in different countries. This long-term effort led to the general use of the "emic" term *evangélicos* to refer to Latin American Protestant believers, for the most part converted from Catholicism. Moreover, the traditional Pentecostal churches, and the newer Neopentecostal and Neoapostolic megachurches, inherited many of the features of Latin American traditional Evangelicalism, such as its missiology, centrality of the Bible, theology of the Cross, eschatology, evangelism, and activism, and for that reason these churches are also classified as *Evangelical* churches (Gladwin 2019), the term that will be used throughout this article. The hegemonic Catholic church considered these Evangelical groups as sects that were stealing its members with an imported faith. In many countries, religious control was the norm, since Catholic authorities were quite influential in governmental decisions, making it very difficult for Evangelical groups to operate freely, obtain legal status, own properties, have recognized seminaries, have access to mass media, hold public crusades, and many other restrictions. As Todd Hartch (2014, p. 22) has described, Evangelicals in Latin America understood that they were members of a religious minority that had endured "social stigmatization and violent persecution". Despite the fierce resistance of the Catholic bishops, Evangelical churches

kept on expanding, especially among the poor in the immense Latin American metropolitan centers that developed between 1950 and 1980 when urban population reached almost 80% in many countries.

During a good part of the 20th century, these small congregations represented a somewhat secure refuge from the anomie produced by large metropolitan centers, and they became the main reproductive instrument for the initial Evangelical growth (Semán 2019). Christian Lalive D'Epinay (1968) believed that the small Evangelical congregation was a protective environment for rural migrants searching for work and better living conditions in the megacity. For Lalive D'Epinay, the Evangelical church reproduced the structures of solidarity left behind in the rural fields, helping migrants bear the burden of city life, extreme poverty, homesickness, low-skilled jobs, and governmental indifference. This close-knit community provided a survival social network, mimicking the paternalistic model of an extended rural family, where the pastor served as a father-like figure, much like the farm landlord in their rural home. The church was seen as a radical *substitute society*, separating individuals from active integration and participation in civil society. At the time it was a very important proposal, which confronted the prevalent Marxist idea, supported by Catholic bishops, of an exogenous plan to convert Latin America to Protestantism, and therefore making it subject to the possibility of North American political control. D'Epinay's thesis emphasized an endogenous origin to Evangelical church multiplication, in large measure free of the control of foreign missionaries, although it was not sufficiently complete to accommodate the successive mutations that these churches underwent in the following years (Bastian 2008).

Using a Weberian approach, Emilio Willems (1967) considered that Evangelical churches tended to be more democratic and egalitarian, without the need for a clerical elite to maintain paternalism. Conversion to Christ meant starting a process of sanctification within a church community that provided discipline, strict moral standards, psychological and economic security, and training to sustain life in megacities with dignity. Willems considered the Evangelical church as a *compensatory society* that affirmed believers to become social actors in a society in transition. In both cases, social anomie acted as a catalyst for the search of a new religious community, with a new social network, and a new world vision and norms that made urban life more bearable.

Many migrant, displaced, and marginalized populations affected by the cyclic Latin American crises converted to Evangelical Christianity, joining its churches and becoming active agents for future expansion and church-planting efforts. However, these theories did not adequately explain why these Evangelical believers left behind their social and political passivity of those earlier communities studied by D'Epinay and Willems, to engage in the national political arena. Mansilla (2014) criticized D'Epinay's idealized paternalistic model of the *hacienda* or rural farm for the relations established in the churches in the periphery of the cities, basically because it was based on exploitation, inequality, and masculine roles, while the Evangelical communities showed solidarity in the midst of poverty, egalitarianism, and an active participation of women in their expansion. Moreover, despite the size of the communities there was also the permanent hope to gain nations for Christ, which activated their compassion, missional zeal, and their vision for healing and growth, which eventually would be expressed in how they lived their beliefs in the face of crisis and extreme social conflict. Through the years, Evangelical communities would try to avoid the escapist rhetoric or apocalyptic discourses in order to address real-life issues such as addictions, violence, family conflicts, inadequate housing, homelessness and others, by making these aspects of their everyday life part of their "religious quest" (Rubin et al. 2014).

More recent explanations explored the changes in religious plurality in the region, particularly towards the end of the 20th century, as a contributing factor (Burity 2020). Gill (1999) offers a *supply-side explanation* to Latin American Evangelical growth, in which religious diversity and affiliation increased when regulations favoring Catholicism were progressively loosened. Having the monopoly of belief did not mean that the whole pop-

ulation was adequately served; in fact, poor areas, especially the new *barrios* and *favelas*, did not have enough parishes, much less clergy presence on a regular basis.[1] Evangelical churches came to fill the gap, providing religious goods and services required by the poor populations—a phenomenon seen as a form of *patched Catholicism*, where Pentecostal churches recreated the traditional popular patronal festivities with new liturgical expressions, messages, and symbols (Bastián 1994, p. 183). An interesting example of the supply-side dynamic of growth occurred in Argentina in 1954 during the 58-day crusade of evangelist Tommy Hicks, which was favored by the government of Juan Domingo Perón, who deliberately opposed the Catholic hierarchy to facilitate the event (Wynarczyk 2021). Caravans of sick people came searching for miracles from suburbs and other regions as well as from neighboring countries. Authorities estimated that six million people attended the crusade, with over 400,000 people gathered on the last day alone. By the opening of this small crack in the ironclad monopoly of faith of the Catholic church, Argentine Evangelicals discovered that they could reach multitudes that needed what they could offer, giving rise to a new evangelistic fervor and the emergence of the leadership that was going to be responsible for the consolidation and growth of the Argentine Evangelical church in the following decades (Zielicke 2012). The model was copied by other Latin American evangelists such as the Argentine evangelists Omar Cabrera, Carlos Anacondia, Héctor Giménez (Wynarczyk 1989), Luis Palau (Hartch 2014), Alberto Motessi, and the Puerto Ricans José Joaquín "Yiye" Ávila, Jorge Raskie, and Eugenio Jiménez Rivera, and many others from all over the continent, who travelled to many cities spreading the message of the gospel and meeting with politicians and authorities wherever they went (Zaldivar 2015).

The example of the previous paragraph shows the effect that reducing legal barriers or softening regulations to new religious movements has in fostering diversity, increasing religious mobilization, and inducing church growth (Finke 1990). More broadly, the more systematic—although nonuniform—process of deregulation started in the region during the 1970s, prompting the growth of the Evangelical population from 4% in 1970 (the break point to religious instability) to 19% in 2014 (with a fast-growing slope), with a corresponding drastic reduction in the Catholic population to 69% (PRC 2014). Numerical changes that demonstrate the large proportion of religious transformations in the region characterized by an expansive Evangelical church (Pérez Guadalupe 2017), and the loss of control by the Catholic Church (Bastián 1994, p. 179), anticipating even more transformations in the following decades.

Knowing that in Latin America most of the Evangelical growth still comes through small, self-reproducing, Pentecostal congregations located in the peripheries of megacities (Semán 2009, 2019), many of them founded and pastored by women (Méndez 2009), it is worth asking: What has been the role of megachurches in the expansion of Protestantism in the region? What is the real impact of the megachurch model and its leaders in Latin American society? And what is the future of this ecclesiological model in the region? One of the most important changes introduced by megachurches was the shift of their missiological target towards the middle and upper classes of Latin American society, where the Catholic church was very strong. This missiological mutation implied drastic changes in discourse, aesthetics, style, methods, and social participation of churches. Somehow, the fierce and pious Evangelical churches, led by uneducated leaders, with extravagant rituals and practices, became acceptable to the affluent and educated social classes. The movement from what the Catholic bishops stigmatized as an invasion of *sects* towards a more respectful status of a *church* had been started (Koehrsen 2017). The cracks that opened in the monopoly of the Catholic church in Latin American society were soon followed by a weakening of its hegemonic presence and the subsequent migration of its middle-class members to the new megachurches, which offered attractive programs and channels of participation contextualized to these socioeconomic sectors of Latin American society (Bravo 2020).

Evangelical megachurches were able to learn techniques to identify the customer's needs and desires, and to innovate in the provision of novel programs and services that

could reach other sectors of Latin American society, particularly the younger generations. This paradigmatic shift required that churches and ministries had the liberty to adapt and experiment with new forms of worship, preaching narratives, theology, and ecclesiology. To be effective in evangelizing their new missional targets, successful megachurches also had to release themselves from their denominational straitjackets. Pablo Deiros, who led the transformation of a historic Baptist church founded in 1883 by Swiss missionaries into the megachurch *Iglesia del Centro* in Buenos Aires (Hong 2011), stated that denominationalism "was in terminal state and deserved a decent burial" (Deiros and Wagner 1999, pp. 52–53). Moreover, the era of foreign mission control also ended, which meant that a new age of indigenous independent nondenominational churches led by autocratic, entrepreneurial pastors, loosely connected through rapidly spreading leadership and church networks was started. Referring to the Brazilian case, Paul Freston stated that "Protestantism (was) *national, popular* and *rapidly expanding*" (Freston 2001, p. 11) and that newly arrived foreign churches no longer created an impact in the country.

## 3. Methodology

A total of 131 published works directly related to megachurches in Latin America were considered for this review (Additionally, 70 other references serve to provide context or background, or as theoretical support), 91 of them since 2010. The search engines of SciELO (Scientific Electronic Library Online); Redalyc (Red de Revistas Científicas de América Latina y el Caribe, España y Portugal-Network of Scientific Journals of Latin American and the Caribbean, Spain and Portugal); and Google Scholar were employed, as well as sets of bibliographic data obtained by the author in three previous projects on cell groups, prophecy, and worship. Although preference was given to articles in peer-reviewed journals in Latin America and elsewhere (60 articles), especially those published in Spanish and Portuguese, books (29), book chapters (18), reports (3), congress presentations (2), doctoral (9) and master's thesis (2), blog reports (1), and news websites (7) were also considered in the analysis. Undergraduate theses, very common in Latin American universities, were not cited, although many of them addressing issues related to megachurches were obtained and filed. The bibliographic research focused on Spanish-speaking Latin America, although references to some Brazilian megachurches that have strong links in the region are made whenever necessary. The topic of Brazilian megachurches would require a complete review article of its own, and for that reason I am deliberately leaving it aside for other researchers to undertake such an endeavor. A considerable amount of research has been conducted regarding the extraordinary growth of the Brazilian Pentecostal *Igreja Universal do Reino de Deus (IURD)* in Spanish-speaking Latin America, which tends to overshadow other Evangelical megachurches in the region; for that reason, these publications were not considered in the review.

The search for these references was based on a contextualized modification of the classification proposed by Bauman (2022), which considered several topics concerning the global study of megachurches, such as *Geographical focus* (country or regional emphasis), *Growth* (numerical expansion, theories, methods), *Worship* (liturgical changes, use of contemporary music, technology), *Leadership* (models, theories, abuses), *Demographics/Social Context* (generational changes, growth of pluralism, globalization), *Gender and Sexuality* (women in ministry, theological influences, cultural changes), *Theological Orientation* (healing/deliverance, prophecy, spiritual warfare, prosperity, dominion, mission, social gospel, eschatology, kingdom theology), *Activities and Practices* (small groups, crusades, conferences, large gatherings, media ministries, social services), *Administration/Marketing* (business models, branding, franchising, social media, commodification, organizational models, networking), and *Political/Social Processes* (political participation, public image, governmental influence/interference, regional issues). This classification allowed one to observe in which areas the research efforts have been stronger over time, and which have been neglected. For example, publications in the *Political/Social Processes* category have been growing since 2016, becoming the most important of those considered for this paper.

As we will see in more detail later, this demonstrates the growing interest of researchers on the influence of megachurches in political processes in the region, and regional journals dedicating special issues on the topic.[2] On the contrary, the *Worship* category has very few publications, which shows that despite the popularity and the big changes produced by contemporary worship music in the Neopentecostal megachurches in the region, there is an absence of research about worship theology, historical developments of church music, cultural influences, and technological advances.

Once the research papers were collected, coded, and classified, it was decided to conduct an integrative review (Snyder 2019) of their content and importance according to three broad catalysts to the emergence of megachurches in Latin America: *Church Growth Methods*, *Pentecostalization*, and *Theologies of Power*. One of the practical reasons to integrate the papers in the three broad categories was that, because of the lack of research, some of the above-mentioned Bauman's categories had very few articles to work with. The three proposed categories have enough descriptive power for an initial characterization of the field and to analyze the potential gaps. The definition of each one of the catalysts is discussed in the following section, and the literature regarding each one of them is analyzed. These catalysts were chosen because they integrate some of the explanations to the extraordinary growth of Evangelical/Pentecostal churches in the region in the latter part of the 20th century and especially in the 21st century. This integrative review approach, although requiring a more demanding literature search (Torraco 2016), allows for the assessment and critique of the published material in order to see many aspects that have been previously overlooked, to propose new areas of research, and to uncover the contributions of Latin American megachurches to their counterparts in the Global North. For example, while small groups are considered as important tools in seeker-sensitive megachurches in the Global North, they have a completely different intrinsic value in their implementation within Latin American megachurches. Similarly, the way that theologies of power are regarded by Latin American Evangelicals, who have been a powerless minority until recently, has a whole different connotation than for USA churches, who seek to regain their traditional influence over government and society.

## 4. Catalysts for the Emergence of Megachurches in Latin America

### 4.1. Church Growth (CG) Methods

Better-known in Spanish as *Iglecrecimiento* (*Iglesia*-Church, *Crecimiento*-Growth), it was originally presented as a scientific methodology to measure the numerical growth of a congregation and to diagnose the reasons for its stagnation or slow growth rate. However, in practice, *Iglecrecimiento* became a set of empirical guidelines that helped church leaders deal with the intricacies of the different elements that affect church growth, which were widely popularized and tested in Latin America through schools and conferences of church growth in the last two decades of the 20th century.

The church growth movement gained notoriety at the School of World Mission (SWM) of the Fuller Theological Seminary (Pasadena, California), under the guidance of its founder Donald McGavran, and later by C. Peter Wagner. McGravan was a missionary in India, where he formulated his initial theories by observing the obstacles that the caste system imposed on evangelism (McGavran 1970). Wagner also spent many years in Bolivia, describing in a book his observations on the extraordinary growth of Pentecostalism in Latin America (Wagner 1973). Other currents that added refinements to CG also became popular in Latin America, such as Christian Schwartz's *Natural Church Development*, which emphasized church health (Rämunddal 2014). CG theory and practice focused on proposing hypotheses about the causes of congregational growth, articulating the results in a scientific language, such that, by characterizing and classifying church growth, interventions and optimizations could be proposed. Four essential principles characterize CG methods (Watson and Scalen 2008): (1) *Quantitative measures* of success (attendance, economic income from tithes and offerings, number of new believers, number of cell groups); (2) Adaptation of message and style according to the context of *homogeneous social groups*

(HSG) or homogeneous units whose sociological characteristics have to be determined. The HSG states that: *people prefer not to have to cross social, racial, or linguistic boundaries to convert to the gospel*. It is one of the most notorious postulates of church growth, introduced by Donald McGravan and popularized by C. Peter Wagner (Wagner 1978); (3) Application of contemporary *marketing techniques and organizational structures* that promote efficiency; (4) Fostering the value of working through *networks* of churches and leaders.

During the 1970s and 1980s, CG experts investigated the reasons for the extraordinary numerical expansion of the Latin American churches, trying to understand the ecclesiological models to see what could be systematized and reproduced in the North American context. Paradoxically, some ideas initiated in churches in the Global South were theoretically and methodologically refined by these researchers who ended up producing the novel CG literature (Bialecki 2015). One of these successful models was the *cell structure*. Experts in quantitative CG methods discovered and designed new tools and procedures, based on the cell group structure developed by David Cho in Korea, that could be offered to those congregations that had stagnated in their growth. The cell model served to affirm the HSG principle, where cell groups represented those homogeneous units of the larger heterogeneous body that was the megachurch, where thousands of those smaller communities could interconnect.

After the 1973 10th World Pentecostal Congress celebrated in Seoul, and Cho's speaking tour in Latin America in 1984, churches were eager to adopt the Korean model of cell churches with the help of experts from SWM and Cho's own CG institute. Churches and denominations around the world were amazed by the unlikely growth of the *Yoido Full Gospel Church* (YFGC) in a country that was not traditionally Christian. Many Latin American pastors traveled to Korea or attended regional workshops where the new concepts and methods were presented, ending up copying or adapting the model, with an important numerical growth in a short span of time (Comiskey 1997). One of the outstanding examples is the indigenous *Misión Cristiana Elim* of El Salvador (Danielson 2013), which is currently ranked as the largest church in Latin America and the third in the world (Roberts and Yamane 2016, p. 324), with a growing presence in the USA among Hispanics, due to the large Salvadorian migration (Danielson and Vega 2014). Founded in 1977, Elim has an estimate of 10,000 cell groups that meet every Saturday, but the church also holds meetings every day of the week and six on Sunday at their central location. Elim holds a traditional Pentecostal theology, including the baptism of the Holy Spirit with the sign of glossolalia and the belief in healing, with very conservative approaches to life, in a way resembling those close-knit sectarian substitute societies observed by Lalive D'Epinay. Somewhat contrary to the trend of Latin American megachurch leaders, Mario Vega, Elim's current pastor, has been moving towards a more progressive theological stand. In this regard, he has sought the help of groups such as *Fraternidad Teológica Latinoamericana* (FTL)[3] in order to apply holistic mission principles (*Misión Integral*) (Clawson 2012) to complex issues of Salvadorian society, such as "political corruption, environmental issues, joblessness, and gang violence" (Wadkins 2018). Perhaps troubled by the paradox of such a large Evangelical growth—about 36% of the national population (Christian et al. 2015)—and the scarce social transformation that is observed in the country, Vega is conscious of the role of megachurches not to remain indifferent, but to influence and promote changes (Vega 2013, p. 54).

*Fraternidad Cristiana*, founded in 1979, was the first congregation in Guatemala to implement the Korean cell church model (Reu 2019), and now claims to have over 15,000 members. Tobias Reu sees that the cell structure contributes to the formation of a leadership mentality in the members, because "the system requires the steady recruitment of volunteers to minister to domestic mini-congregations" (p. 751), who are trained in leadership skills and challenged to move up the ladder to roles such as area supervisors. The training fosters a *multiplication effect* that CG requires, producing leaders that multiply themselves. This urge to make cells grow and multiply will be seen in all cell-based megachurches in Latin America. Church size, expressed in the number of cells, "is an indication that

they are doing something right" (Garrard 2021, p. 204), that the congregation has God's favor. Moreover, by staying close to the realities of the neighborhoods where they live, cell leaders "take the gospel to the people, to lead by example, to induce others to become leaders, and to be a light in the darkness" (Reu 2019, p. 757), in a society where crime and violence abounds, and where families struggle to survive poverty, under the indifference and neglect of politicians and governmental officials.

By the end of the 1980s, a large majority of cell churches in the region followed Cho's system or its variations. Some, although not the majority, started to combine the classical worship service and program-based model with cell groups, termed by SWM experts as the *metachurch* (George 1991), where cells were just another program and did not involve most of the members. Others began to look for alternative methods. This was the case of Colombian pastors Cesar and Claudia Castellanos of *Misión Carismática Internacional* (MCI) at Bogotá (Colombia), who had implemented the Korean model in 1986 but were seeing very slow progress. By 1991, they had managed to establish some 70 cells in five years, so they progressively began to tweak Cho's classical model to accelerate growth. The results were immediate, with 1200 cells in 1994, reaching about 45,000 cells by 2001 (Comiskey 1999). Besides MCI, many Colombian churches adopted MCI's model (Beltrán 2012), now known as the *G-12* system. Among them were *Manantial de Vida Eterna* (Bogotá), *Centro Cristiano Internacional* (Cúcuta), *Misión Paz a las Naciones* (Cali), *Centro Bíblico Internacional* (Barranquilla), *Misión Carismática al Mundo* (Cali), and *Sin Muros Ministerio Internacional* (Bogotá).

Thousands of churches in Latin America and the rest of the world adopted the G-12 model. For example, at London's megachurch *Kensington Temple*, G-12 implementation began in its small Hispanic church, which grew very quickly, prompting the implementation of the model to the whole congregation (Kay 2006). Likewise, many Brazilian churches massively welcomed the model. However, due to MCI's franchising scheme used to distribute the model, many congregations tried it at first, but along the way they decided to abandon or adapt it. To ensure trademark uniformity, churches were required to establish a covenant relationship with MCI; become certified to be able to use the written materials; implement all the steps in the training path; follow all the recommendations for the conformation of the groups (size, homogeneity, etc.); continue to use the number twelve as a symbol of identification with the original model; and also participate in the activities programmed by the network, such as conferences and/or trainings (Comiskey 2014). The logical and expected result, quantifiable and guaranteed by the franchiser, was an accelerated growth of the church. Organizational sophistication is one of the aspects that becomes evident in these megachurches, by incorporating training curricula (such as the *G-12 Escuela de Liderazgo*, or the Fraternidad Cristiana *Facultad Teológica*), coaching, and tight accountability of the leaders, some sort of McDonaldized coordination and control can be implemented (Mora 2022a). Church interrelations based on the G-12 franchising system have been fundamental to the development of very large church networks in Latin America, where other religious goods and ideas are also exchanged, as recent research among Baptist churches in Argentina has shown (Marzilli 2019, p. 259). Nevertheless, many churches resisted to pay license fees, made minor modifications, or changed their nomenclature, but kept the essence of the model, even launching their own brand, such as Abe Huber's MDA (*Apostolic Discipleship Movement*) in Santarém (Brazil), whose *Igreja da Paz* megachurch has approximately 65,000 members, representing 22% of the city population and 87% of the city's Evangelical community (Brevi de Moura and Ordones 2018). *Igreja da Paz* is currently the second-largest church in Latin America after *Misión Elim* (Roberts and Yamane 2016).

At a symbolic level, the trend towards internationalization and globalization is evident in the names given to G-12 churches such as the *Misión Carismática Internacional* of Bogotá, the *Mundial do Senhor Jesus Cristo* pastored by Valnice Milhomens in Brazil, or the *Movimiento Carismático Internacional* of Lima-Perú (Santos Andrade 2010). Not all G-12 churches (or disaffiliated but maintaining a similar model) can be considered megachurches, but several are among the largest in Latin America, such as *Casa de Dios* in Guatemala, led by apostle

Cash Luna (Gomes 2021), where a modified D-12 system was implemented. The size and active presence of the churches through the cell groups in neighborhoods give them extraordinary convening power and influence, which can be exploited at a political level. This is the case of *Igreja Batista de Lagoinha* in Minas Gerais, the fourth-largest in Latin America, where one of its main leaders is Damares Alves, Minister of Women, Family, and Human Rights of Jair Bolsonaro's government. Through the exploitation of symbolic and mystical elements, the managerial activity of the pastor of a cell church is equated with that of an apostle, prophet, or patriarch (or matriarch), who possesses a special revelation for the revival and transformation of his nation and the world. For the senior leader of the megachurch, cell growth and continued leadership training will eventually lead to reach a city, an entire country, and other nations, through a well-trained army of leaders and members under his/her covering.

Even though Rich Warren's book, *The Purpose Driven Church* (Einstein 2008), became very popular in the region after the publication of its Spanish translation, followed by many regional workshops and conferences (Madambashi 2011), it is more difficult to observe the influence of the *seeker-sensitive* methodologies of church growth in Latin America. Some churches adopted the model with enthusiasm (Corpeño 2011); however, the Pentecostal background of many pastors, apostles, leaders, and church planters made them rely more on other spirit-filled or pneumatological dimensions of growth. Nevertheless, some researchers associate the influence of Warren and others on the development of the managerial aspect of Latin American megachurches. Some sort of translation of the management and coaching mindset, with its continuous demands on training and professionalization, became part of church life (Schäfer et al. 2013).

More recently, the term *post-denominational* (PD) has been proposed to refer to churches that are congregating Christians dissatisfied with traditional Pentecostalism and Neopentecostalism (Ibarra and Gomes 2022). In a way, this categorization is similar to what Donald Miller described as *new paradigm churches* (NPC) (Miller 1999), a type of churches that are more culturally engaged, and where styles—musical, communicational, or esthetic—dress codes, and ambiance are more attractive to younger generations, relying much more on technological means to reach their audience (Ibarra 2019). PD megachurches propose a different model of Christianity that is adapted to the cultural understanding "about choice, individualism, autonomy, the importance of the self, therapeutic sensibilities" (Sargeant 2000, p. 31). In these churches, the problem of sin is minimized, strict rules disappear, focusing on attracting people to church, using all the resources and techniques available. This entails building a creative core of professional people capable of innovating in the way the Gospel is communicated, deconstructing the Gospel message to make its proclamation more relevant and pleasant and creating a satisfying experience during church meetings (Ibarra and Gomes 2022).

Due to the heavy evangelistic and missional emphasis of Neopentecostal churches, many of them identified with new paradigm churches such as Calvary Chapel and Vineyard, using relaxed attires, contemporary music, and a language that was appealing to the younger generations. However, this trend did not include a more in-depth deconstruction of the Gospel message. For example, the Brazilian *Bola de Neve Church*, started in the year 2000, which appeals to younger surfers and skaters of São Paulo's suburban area, at first glance could be classified as PD. Nevertheless, despite the use by its founder, *Apostle Rina* (Rinaldo Luís de Seixas Pereira), of younger generations' slang in his messages, and the group having symbols that identify the members with the Brazilian beach and sports subculture, as well as a musical style and branding that tries to bring a flavor of a relaxed youth culture, including Carnival parades using the traditional *batucada brasileira* and sharing the performances in social media (Oosterbaan and Godoy 2020), the structure and theology of the church are quite traditional and legalistic (Pinto Ribeiro and Da Silva Cunha 2012).

Few PD churches in Latin America are megachurches in the classical numerical sense, but many are in the virtual space that the social, mobile, analytical and cloud technology allows. Ibarra (2019, 2021), Ibarra and Gomes (2021, 2022), have studied some of these

churches in Northern Mexico. *Ancla* church in Tijuana (Baja California), founded in 2015, is a megachurch of over 5000 young members that relies on a creative team that exploits visual arts and graphic design for the worship services and the positioning of the brand through social networks. Edson Gomes (2020) studied *Más Vida Guadalajara*, a *multi-site* church that evolved from a Neopentecostal congregation founded in 1983 and that currently has a Sunday attendance of over 1500 people in its central campus in Morelia, and many more in other satellite sites such as Guadalajara, Toluca, Querétaro, León, Ciudad de México, Acapulco, León, Dallas, Orlando, and Caracas. Both churches, *Ancla* and *Mas Vida*, have strong links with other megachurches in the USA that follow similar practices and methodologies. These churches develop advanced branding strategies and digital *customer relationship management* (CRM) methods to position themselves in the market and follow up with their members (Gomes Rego 2020). This is a technology-boosted way of applying the homogenous unit principle to reach younger middle-class tribes that reject any identification with traditional Evangelical Christianity. It is difficult to determine if the main growth of these churches comes from new converts or from second- or third-generation Latin American Pentecostals, that are inserted in current alternative subcultural movements in major cities of the continent (Espinosa Zepeda 2017), whose aesthetics, musical tastes, irony, critique of traditional Christianity, and new hermeneutics of the Gospel turn out to be too difficult to understand or accept for traditional churches (Wilder and Rehwaldt-Alexander 2010).

The recent spread in the region of global brands based on the megachurch model needs to be considered regarding the spread of PD churches in Latin America. For example, the Los Angeles-based *Mosaic Church*, considered one of the most innovative churches in the USA (Marti 2005), started a new branch in Mexico City in 2018, which is growing very quickly, and another one in Quito (Ecuador) in 2020. Hillsong started a church in São Paulo (Brazil) in 2016 with over 3000 people in attendance. Also, another Hillsong congregation was opened in Buenos Aires in 2017, following the same methodology as in Brazil, quickly reaching about 4000 people attending every Sunday (Infobae 2017). It has been shown that Hillsong prefers to carry out church planting in urban centers (*global cities*) that are influential, with high-tech resources, that serve as hubs for "highly skilled labor, cultural elites, expats, migrants, and tourists" (Klaver 2018), such as professional Brazilian migrants in Sidney (Rocha 2016, 2017). *Ancla*, *Mas Vida*, Hillsong, and Mosaic churches tend to attract millennial believers that are part of the creative and technology industries, who are transitioning from Neopentecostalism to this new style of church (Klaver 2018). Rocha has recently coined the term *Cool Christianity* (Rocha 2021) to refer to a "*fashion-celebrity-megachurch industrial complex* that makes Christianity attractive to middle-class youth who do not find a home in more conservative churches". This new kind of congregations that use the "novelty, transgressive force, and emotional edge of the religious experiences" (p. 132) to attract globalized postdenominational young urbanites seems to be the *fourth wave of Brazilian Pentecostalism*[4] (Fernandes 2022). It would be interesting to determine if this trend is a generational issue, in which millennials and generation Z believers do not identify with the faith of their elders and embrace more open theologies and less predictable churches, or if it is a contemporary version of the fascination of Latinos with the "culture and lifestyles of the global North" (Rocha 2017, p. 126).

### 4.2. Pentecostalization

There is a worldwide process of *Pentecostalization of Christianity*, including the Catholic Church.[5] In Latin America, this became more obvious after the 2006 Pew Research Center study (PRC 2006), later confirmed even further by PRC in a follow-up study (PRC 2014), which showed that at least one in every four Latin Americans were Christian believers that had experienced the charismata of the Spirit (speaking in tongues, divine healing, prophesying, deliverance) and who were involved in lively and exciting churches where they could nurture their personal faith and express their missionary agency. Due to its expansive nature and mobility, a Pentecostalized form of Christianity has become

globalized. The Pentecostalization of the church was identified very early by CG researchers at SWM who thought that it would be an important improvement to McGravan's original pragmatic theories and methods (Cook 2000). SWM Scholars tried to identify which elements of Pentecostalism were conducive to explosive numerical growth, such that other groups, regardless of their denomination or theological orientation, could implement them. During the 1980s, pragmatism met revival, and spiritual power anointed technical knowledge, such that churches could be awakened and grow. Following Charles Finey's idea that revivals needed to be "worked up" and promoted (Johnson 1969, p. 353), a global search for Pentecostalized supernatural CG methods was initiated, providing an impressive new set of tools that could be offered to facilitate growth (Bialecki 2015, p. 181).

Latin America proved to be very fertile for this search and became a test ground for the new ideas that came up (Swartz 2020). At a time when the YFGC in Korea was growing very fast, some Argentinian megachurches called the attention of the experts. Wynarczyk (1989) studied the beginnings of the megachurch *Visión de Futuro* founded by Omar Cabrera (father) (see Table 1). The congregation, currently led by Omar Cabrera Jr., is still quite strong and growing, having evolved from classical Pentecostalism to Neopentecostalism, and lately as part of the New Apostolic Reformation (NAR) (Koehrsen 2016). CG experts were particularly interested in the intuitive forms of *power evangelism* and *spiritual warfare* that Omar and Marfa Cabrera cultivated in their ministry (Wynarczyk 1989, pp. 23–25). C. Peter Wagner visited Cabrera in Argentina in 1985 to study the model (Holvast 2009, p. 55), later calling him the *Chief Apostle* of the Argentine revival (Wagner 1998, p. 21). He even acknowledged that Cabrera inspired him with the idea of the direct relationship between strategic spiritual warfare and church planting.

**Table 1.** Largest megachurches in Latin America (modified from Leadership Network list).

| Sunday Attendance | Name of Church | City-Country | Theological Orientation | Date Founded |
| --- | --- | --- | --- | --- |
| 50,000 | *Misión Cristiana Elim* | San Salvador, El Salvador | Pentecostal | 1977 |
| 50,000 | *Primera Iglesia Metodista Pentecostal* | Santiago, Chile | Pentecostal | 1911 |
| 50,000 | *Igreja da Paz* | Santarém, Brasil | Neopentecostal | 1993 |
| 50,000 | *Batista da Lagoinha* | Belo Horizonte, Brasil | Neopentecostal | 1957 |
| 42,000 | *Comunidad Cristiana Agua Viva* | Lima, Perú | Neopentecostal | 1985 |
| 26,545 | *El Lugar de Su Presencia* | Bogotá, Colombia | Neopentecostal | 1993 |
| 25,000 | *Misión Carismatica Internacional* | Bogotá, Colombia | Neopentecostal | 1983 |
| 25,000 | *Lluvias de Gracia* | Ciudad de Guatemala, Guatemala | Neopentecostal | 1991 |
| 25,000 | *Ministerio La Cosecha* | San Pedro Sula, Honduras | Pentecostal | 1977 |
| 20,000 | *Fraternidad Cristiana* | Ciudad de Guatemala, Guatemala | Pentecostal | 1979 |
| 20,000 | *Casa de Dios* | Ciudad de Guatemala, Guatemala | Neopentecostal | 1994 |
| 20,000 | *Visión de Futuro Church* | Santa Fe, Argentina | Neopentecostal | 1972 |
| 20,000 | *Ondas de Amor y Paz* | Buenos Aires, Argentina | Neopentecostal | 1983 |
| 20,000 | *Misión Paz a las Naciones* | Cali, Colombia | Pentecostal | 1999 |
| 15,000 | *Rey de Reyes* | Buenos Aires, Argentina | Neopentecostal | 1986 |
| 12,000 | *Comunidad Apostólica Hosana* | Ciudad de Panamá, Panamá | Pentecostal | 1980 |
| 12,000 | *Movimiento Carismático Internacional* | Lima, Perú | Neopentecostal | 1998 |
| 10,000 | *Centro Familiar de Adoración* | Asunción, Paraguay | Pentecostal | 1985 |
| 10,000 | *Tabernáculo Bíblico Bautista* | San Salvador, El Salvador | Fundamentalist Baptist | 1977 |
| 10,000 | *Más Vida* | Morelia, México | Neopentecostal Postdenominational | 1984 |

Similarly, through the public crusades started by Carlos Anacondia in Buenos Aires, shortly after the end of the Malvinas war, deliverance ministries became very popular, first in Latin America, and subsequently in the rest of the world (Marostica 2011). Pablo Bottari, a leader of the megachurch *Iglesia del Centro*, became director of Anacondia's healing and deliverance tent during Anacondia's massive crusades (Brown 2006). Bottari compiled and systematized deliverance manuals based on the observation of Anacondia's practices, and from his own learning while running the tent (Bottari 1998). Rationalization of these procedures facilitated their dissemination all over the world, becoming an important influence on other procedures and programs put together by megachurches, such as deliverance rituals of G-12 franchise megachurches, *Catch the Fire soaking* prayer, or *Bethel Church Sozo* inner healing program (Weaver 2015). Today, Bottari's deliverance manuals, or versions of them, can be found on countless Christian bookstores around the world, showing the popularity of Pentecostal deliverance ministries in the Evangelical church at large.

Following the footsteps of Carlos Anacondia, around 1992, Claudio Freidzon, an Assemblies of God pastor, led a new wave of spiritual power that manifested through worship, unusual spiritual manifestations, and miraculous healings during the services at the *Rey de Reyes* church in the Belgrano neighborhood of Buenos Aires (see Table 1). Although at the time the church had over 2000 members, Freidzon was restless and longing for a more intimate relationship with the Holy Spirit (Freidzon 1996). After a visit to Benny Hinn at the *Orlando Christian Center* (Florida-USA), many spiritual manifestations started at *Rey de Reyes*. Suddenly, the power of the Holy Spirit started to move among the Argentine middle class. As in those crusades in stadiums, the expectation for healing and deliverance in each meeting remained, but the places, the audience, the aesthetics, and the music were more acceptable to the professional middle-class strata. The movement, coined as *La Unción*, triggered a wave of renewal and revival in the country and beyond that lasted several years (De Seixas Andrade and Raimondo 2016). *La Unción* was fundamental for the initiation of several revivals that occurred in the latter part of the 20th century, such as the Toronto Blessing and its ramifications in the UK, and Pensacola in the USA (Deiros 2018, p. 1039). Algranti studied *Rey de Reyes* as a prototype of a middle-class Latin American Neopentecostal megachurch. Through ethnographic research over several years, many aspects of the life of the church were considered, especially its transit from a classical Pentecostal congregation to its radical move to a less-evangelized area of the city, evolving theologically to a gospel of spiritual healing, prosperity, and warfare (Algranti 2008b). Algranti studied in depth the characteristics of the leadership in the megachurch (Algranti 2005), its implications in Argentine civil society (Algranti 2012), gender relations within the megachurch (Algranti 2007), as well as the practices of healing and deliverance (Algranti 2008a).

Another area that CG experts considered fundamental was the discovery of a sophisticated prayer method called *Strategic Level Spiritual Warfare* (SLSW) which relays on a technique called *spiritual mapping* (SM) (Holvast 2009). In the 1980s, it was proposed that the reasons that churches were not growing had to do with demonic opposition rooted in the sinfulness of the cities and territories, or in the wicked spiritual inheritance of people groups. Therefore, it was necessary to identify those sources of evil through SM and then pray them away through intense periods of worship, intercession, fasting, and publicly naming and rebuking those spiritual forces. One of these prototypes for territorial spiritual deliverance was the highly publicized transformation of Almolonga in Guatemala, which through intense hours of fasting and prayer by local churches overcame poverty and became an agricultural miracle (Garrard 2020). René Holvast (2009) points out that Argentina was the great laboratory where many of the concepts of spiritual warfare were initially tested. Eduardo Silvoso, one of the pioneers of SM, proposed and developed a prototype for the evangelization of any city, based on spiritual warfare techniques, followed by intercession, evangelism, and conquest (Holvast 2009). *Plan Resistencia* sought to take a city for Christ, between 1989–1992, by using innovative spiritual warfare techniques that would cast out the principalities and powers that were stalling Evangelical growth

(Wynarczyk 1995). During the 1990s, the city of Resistencia (El Chaco, Argentina) became the model for the taking of over 200 cities on five continents, making spiritual warfare an essential tool in the CG arsenal (Silvoso 1998). In 1994, Wynarczyk (1995, p. 167) noted the new emphasis, by observing the ministry of Buenos Aires megachurch pastor Héctor Aníbal Giménez, of *Ministerio Ondas de Amor y Paz*. Giménez expanded his deliverance prayer from exorcising demons of illness and personal oppression to spiritual warfare against territorial powers in La Pampa and Río Negro (Argentina), after their SM revealed the genocide of indigenous people that occurred during the XIX century, which was affecting the spiritual, social, and economic atmosphere of those regions. This kind of thinking became so prevalent in all Latin America that it is likely, at some point in their history, that cell groups and regular members in every megachurch in the region engaged in some form of SM/SLSW.

Two Central American churches are at the forefront of using SM/SLSW in their growth practices (Garrard 2020). In Guatemala, Dr. Harold Caballeros, pastor of *El Shaddai* megachurch (O'Neill 2012), has been an active promoter of SM/SLSW as part of the toolkit for church growth, after they discovered that territorial spirits, which derived their power from the Mayan and Aztec worship of the flying serpent divinity of *Quetzalcoátl*, were controlling the area where they were building their church (Caballeros 1999, pp. 29–32). Kevin O'Neill (2010) describes *El Shaddai*'s practice of SM/SLSW when Dr. Caballeros was seeking his nomination for the presidency of Guatemala towards the end of 2006, when selected prayer warriors pledged to pray for 21 days in a row to destroy the principalities and demonic powers that controlled Guatemala City, seeking to produce a social and spiritual environment that favored the launch of Dr. Caballeros' nomination.

In Costa Rica, Ronny Chaves, leader of several apostolic networks and apostle of *Centro de Adoración Mundial*, embraced the spiritual warfare movement from its early days (Carpio Ulloa 2021), becoming an enthusiastic practitioner and innovator.[6] Through SM, Chaves discovered spaces or corridors in Costa Rica where evil forces moved freely to provoke negative influences on God's people. High places in the four cardinal points of the country, and the Catholic *Basílica de Los Ángeles*, were identified as dominated by territorial spirits, and where intense SLSW had to be conducted (Carpio Ulloa 2021, p. 15). These places in Costa Rica have been the subjects of spiritual battles through prayer over the years, but more so as Evangelicals become more involved in politics, showing how SLSW translates into mundane political practice. On one side, megachurches motivate and provide space for continuous intercessory prayer, but on the other, they empower individuals to get actively involved in political actions (Marshall 2016). SLSW became a tool during the 2018 presidential campaign of Evangelical Fabricio Alvarado, when churches launched an offensive of spiritual warfare led by Chaves to "conquer the territory (and) take a position in the places of government, education and the economy" (Murillo 2018). Caballeros and Chaves have presented their SM/SLSW methods in countless conferences of their apostolic networks, serving as a preparation for the introduction of *dominion theology*, as part of Latin American megachurch theological basis.

One last sign of Pentecostalization to be considered is the widespread diffusion of *contemporary worship music* (CWM) and the associated drastic changes in Pentecostal liturgy that developed. Until the 1970s, music in Evangelical churches in Latin America was quite simple, occupying a secondary place in relation to preaching. However, the restorationist wave that started in the 1950s led to the development of a much more sophisticated *Davidic Model* of worship (Perez 2021). This paradigmatic change in liturgy implied a substantial modification of the services, especially with regard to its times and flows, the incorporation of symbols such as banners and flags, and choreographies; rituals were created and recreated of sanctification, purification, sacramental washings, spiritual warfare walks, anointing with oil; and a new freedom was allowed for the expression of various bodily, emotional, and spiritual manifestations such as falling to the ground, laughing, shouting, crying, or dancing in the Spirit. The disruption caused by the innovations introduced by the praise and worship (P&W) movement was described, as early as 1994,

as a true revolution of "the liturgy and the contents of community worship" ([Deiros and Mraida 1994](), pp. 149–51). For many megachurch pastors, the changes in musical style and the new liturgy were excellent catalysts for numerical growth, since they made services colorful and attractive ([Schwarz 2001]()). Contemporary musical styles became popular within churches, and pragmatism became the norm to decide about the theology of the lyrics, the inclusion of some genre, style or musical instrument, or any new artistic form or bodily expression for worship, even the use of stage lighting, display of images, and other digital paraphernalia ([Mansilla 2006]()).

Very little has been published that assesses the extent of these influences throughout the region during the initial years. Important Neopentecostal megachurches in Venezuela (*Iglesia Evangélica Pentecostal Las Acacias*, IEPLA) ([Berryman 1996](), p. 133; [Burch 2016]()), and Colombia (*Comunidad Cristiana de Fe* of Cali), were poles from where the Davidic model was disseminated and multiplied, but these experiences have not been studied yet. Almost simultaneously there was also the emergence of the Latin American P&W movement, led by Marcos Witt and others[7] ([Gladwin 2015]()). Meanwhile, the influence of the Davidic model and restorationism on the CWM mainstream in Latin America was supported in Witt's writings and teachings and through his own record label ([Witt 1993]())[8]. The rise of the well-known and influential Christian record label Integrity/Hosanna Music (IHM) ([Perez 2020]()), which produced many albums in Spanish, also helped to make the P&W movement grow exponentially. In the process, many megachurches began to offer good salaries to worship leaders, apart from providing an impressive physical space, and all the audiovisual equipment necessary to make their own productions, prioritizing popular rhythms and styles, seeking excellence in sound quality, professional artistic execution, and the use of well-studied scenery and lighting ([Bowler and Reagan 2014]()). It should be no surprise that Marcos Witt was pastor and musical director of the Hispanic ministry of *Lakewood Church*, one of the largest megachurches in the USA. Other well-known Mexican worshipers, Coalo and Lorena Zamorano, accompanied Witt in these functions until 2013, when they went on to lead the Hispanic ministry at *Champion Forest Baptist Church*, one of the largest megachurches in Houston. When Witt left in 2012, he was replaced by Danilo Montero, another of the pioneers of the P&W movement in Latin America.

One megachurch that employed CWM as part of its growth strategy is the Brazilian congregation *Renascer em Cristo*, founded by Estevam and Sonia Hernandes ([De Souza 2011]()). Through careful marketing, using various strategies and communication channels to convey their message, such as radio and television, festivals and live concerts, mega-shows in stadiums with internationally recognized bands, production and release of CDs, and video streaming, the church contributed to the creation of a vast market for Christian music in the growing young Evangelical population of the country ([Dolghie 2005]()). The release of its first record label, *Gospel Records*, was in 1989. The worship record series called *Renascer Praise* (more than 18 titles), started being recorded in 1993, obtaining a Grammy nomination in 2014. Along the way, the megachurch signed distribution contracts with companies such as Sony Music in 2010 and Universal Music in 2013. [De Souza]() ([2011](), p. 23) showed that *Renascer em Cristo* developed a cultural product that trespassed the Evangelical segment, gaining space to the secular music market, thus attracting young people to the megachurch.

According to the logic of the musical market, the search for new ways of musical expression, the intensive use of technological resources to perform on public stages, and the constant innovation and creation of new styles are characteristics of worship music in megachurches ([Evans 2017]()), which is only possible due to the financial power of these organizations. This is the case of the *Casa de Dios* church in Guatemala City, which organizes the *Lumination* electronic music event, characterized by high professional quality and complex production, using sound, screens, pyrotechnics, and dance, directed by DJs who lead those gathered into a multisensory show, with the intention of "connecting" the audience with God ([Gomes 2021]()). The use of electronic music, dance, and lighting is an innovation in P&W practices of megachurches ([Moberg 2015]()). The technological deployment, the youth of the performers, the sophistication of the stages, and the apparent freedom of the

attendees to dance and move, speaks of a prosperous and relaxed environment where new artistic forms are appropriated and transformed for the preaching of the gospel to a new generation.

*4.3. Theologies of Power: Prosperity, Wealth Transfer, and Dominion*

As Heuser asserts, there is a theological gap in most of the megachurch reviews conducted so far (Heuser 2021, p. 194). It is fundamental to understand the theological foundations that drive the expansion of Evangelical churches because these theologies validate the discourses, approaches, risks, styles, associations, and investments taken by megachurch pastors and leaders to foster growth and gain power within society. This category integrates those theologies that influence and serve as driving forces to several aspects of megachurch growth that were considered in the previous two categories. In Latin America, traditional Evangelical theology had preached an extreme pietism, where money and possessions were considered as temptations and deceptions of the world (Semán 2001). The newer *theologies of power* led megachurches to think about the here and now, to extend the kingdom of God through competition and the free market, establishing business models whose profits were to be reinvested in the growth of ministries and churches, and in achieving new positions of power in society.

Formulated in the USA out of the *law of faith* teaching of Kenneth Hagin, *prosperity theology* centered initially on physical and emotional health. Hagin had the influence of the *new thought* movement through the teachings of Essek William Kenyon, whose hermeneutics of the gospel was based on the prevailing American cultural values of "pragmatism, individualism and upward social mobility" (Quesada Chaves 2019). Hagin's teachings were later expanded to material prosperity by Kenneth Copeland, who included the laws of *blessing* and of *sowing and reaping* (Bowler 2013). After several additions and refinements, the resulting theological corpus is a combination of verbal declarations with symbolic and ritual practices, whose thematic axes focus on faith, physical and emotional health, material wealth, and personal victory (Heuser 2016).

In Latin American megachurches, the basic formulation of prosperity theology must be expanded to include, on one side, what has been termed as the *transfer of wealth principle*, and on the other, *dominion theology*. For that reason, I will refer to the different combinations of prosperity, dominion, and wealth transfer teachings and principles as *theologies of power*, because over time these have led to the active involvement of churches in the political arena (Pérez Guadalupe 2017). The term "theologies of self" has also been proposed because they focus and emphasize human capacity building (Garrard 2021, p. 196), something that can be considered as a form of human or social capital. In this regard, church *membership* measures the *religious capital* of megachurches, an indication of its "religious positioning, economic power, and political and social visibility" (Beltrán 2003). Discourses of growth, prosperity, increase, power, and dominion are common because there is a need to constantly increase the religious capital of the megachurch. However, as Aguirre (Aguirre 2022) has recently stated, Evangelical megachurches do not engage in a "radical critique of the existing social order and its basic structures, nor [ . . . ] propose a profound transformation"; on the contrary, they tend to reinforce and reproduce the dominant economic and political system. For example, one common strategy by Christian organizations in Latin America is to offer microcredits to believers that become entrepreneurs that get involved in the financial struggle between debtor and lender, framed in the language of supernatural prosperity (Bartel 2021).

The most common observation is that most megachurches use some contextualized version of *prosperity theology*. For example, behind the growth of MCI in Bogotá, there is a prosperity theology using a *ladder of success* as a metaphor of continuous growth, elaborated by César Castellanos in times of national turmoil, violence, and terrorism. Other Colombian megachurches followed the same approach, encouraging economic prosperity, offering salvation by a powerful and rich Christ, enabling the elaboration of a Christology of success and self-empowerment (Jiménez Becerra 2013). The same holds

true in the prosperity formulated by Claudio Freidzon in Argentina, after the end of the dictatorship period and a time when the country suffered economic mismanagement by the democratic governments during the 1980s and 1990s. More recently, the prosperity message of megachurch apostles Cash Luna and Rony Chaves, and other representative figures of Neopentecostalism and megachurches in Latin America, has been studied using a postcolonial approach (Barrantes Montero 2020). An interesting study from the standpoint of the members of MCI in Bogotá showed that despite their precariousness, they give and tithe beyond their real possibilities, making donations with their credit cards and paying high monthly interest rates to finance the activities of the megachurch, as an act of fidelity to God (Bartel 2016b). Another fascinating investigation with a special focus on women showed the approach of megachurch *Rios de Vida* (Cartagena, Colombia) to combat poverty through entrepreneurship, mixing in their message coaching and spiritual warfare as tools to facilitate the startups of their members (Ramos 2021). *Business and spiritual coaching* is described as one of the "more successful products of the church" offered by Neopentecostal entrepreneurial gurus (p. 214).

The *transfer of wealth* is based on an interpretation of Proverbs 13:22b (Avanzini 1989), which states that the wealth of sinners will be inherited by Christians, in a long-term process that would lead the church to access this wealth (Wagner 2015). In other words, immense fortunes, capitals, stock shares, productive companies, real estate, cash money, and cryptocurrencies, which were amassed by entrepreneurs, merchants, companies, and governments, would eventually make a transition of hands from the sinners who generated and accumulated them to the righteous, who would now use them for the mission of bringing the dominion of the kingdom of God on earth. Megachurches have the mission of receiving (stewardship) and multiplying (entrepreneurship) money for the mission, which means, among other things, to play in the field of capitalist speculation through the financial markets, to create innovative ventures, or to look for investment options that produce the best returns on investment (ROI) for the kingdom. This calls for a new breed of *marketplace apostles* (Mora 2022b), formed from a base of neoliberal professional millennials and middle-aged religious entrepreneurs, independent from denominational attachments, many of them oriented to the technology sector (Schäfer 2021).

Gaiya (2015) proposed to differentiate between *centripetal* and *centrifugal* megachurches, depending on how they view their social commitment and the way they transfer this wealth. *Centripetal* megachurches are inward-looking, channeling the bulk of their human and financial resources towards the church, obtained from the monetary contributions of their members, usually tithes and sacrificial offerings. *Centrifugal* megachurches are more outward-looking, investing their resources to influence the social and political environment, according to their vision and spirituality, progressively evolving towards a more deliberate political participation. It is surprising how little churches invest in social projects, dedicating the largest part of their income to centripetal growth. In Colombia, around 54% of the churches did not have any consistent social work, and those that did, engaged mainly in food distribution (60%), education, health, psychosocial care, family conflicts/domestic violence, and job training (Lozano 2008, p. 264). Another study of megachurches in the UK categorized five key areas of involvement, namely "meeting basic material needs; employment; life skills; children and youth work, and neighborliness" (Cartledge et al. 2019, p. 109). Many Latin American megachurches are involved in some of these categories, sometimes with budgets provided by governmental programs, but their business models, their degree of commitment, and the amount of time dedicated to the projects do not allow for long-term sustainability. There are quite important centrifugal megachurches that have had consistent social work for several years that have not been researched yet. For instance, *Iglesia Evangelica Pentecostal Las Acacias* (*IEPLA*), located in a lower middle-class neighborhood in Caracas, close to poor barrios in the nearby hills, opened a community center with different services in the 1990s, which is still functioning. IEPLA had one of the first successful drug rehabilitation programs in Venezuela (*Hogar Vida Nueva*) and supports a theological seminar with majors specializing in social ministries.

According to Miller and Yamamori, IEPLA had "one of the most extensive networks of social ministries that we encountered" (Miller and Yamamori 2007, p. 10). *IEPLA* could be classified as a *Progressive Pentecostal* megachurch whose members "claim to be inspired by the Holy Spirit and the life of Jesus and seek to holistically address the spiritual, physical, and social needs of people in their community" (Miller and Yamamori 2007, p. 2)

Other centrifugal megachurches prefer to enter the political arena, moving from a purely holistic social approach to a more politicized and aggressive public activism. For example, *Iglesia Maranatha* of Valencia (Venezuela) started distributing bowls of soup in poor locations in the country, progressively evolving to participate in presidential and regional elections. Maranatha's pastor and *apostle*, Javier Bertucci (Mora 2022b), was a presidential candidate in 2018 against Nicolás Maduro, obtaining 11% of the votes. Currently, he leads three legally constituted associations in Venezuela: a religious organization, a nonprofit civil association, and a political party. Despite a series of questions and possible commercial scandals, including lawsuits in progress, there is a network of companies that support and supply the projects of these three associations in areas as diverse as oil, agriculture, food imports, health supplies, books and stationery, and construction companies, with offices in several Venezuelan cities and in other countries. The civil association oversees direct social aid and evangelistic outreach through workshops in neighborhoods and low-income sectors. Maranatha megachurch and the associated political party managed to present 4430 candidates in 335 municipalities in the country in the 2019 municipal elections, surpassing the main traditional and seasoned political parties. Their most recent achievement was obtaining four seats in the National Assembly in the legislative elections of December 2020, becoming the second opposition group in the congress. Without an elaborate political theology, Bertucci's Maranatha megachurch has managed to place itself where they can influence governmental decisions, through an apostolic access to the spheres of power.

The increased "politicization of Pentecostalism occurring in megachurches has a theological basis [ . . . ] identified as *Dominion Theology*" (Heuser 2021, p. 190). Dominion Theology takes the mandate given in Genesis 1:26–28 and Matthew 28, to make, as an obligation to the Church to establish Christ's dominion over the earth by "conquering cities, regions, and nations [ . . . ] bringing individuals to Christ, but also changing the legal system, government, education, churches, synagogues, families, and other aspects of human society in accordance with Christian beliefs and moral norms" (Budiselić 2015). *Dominion Theology* (DT) is very important to understand megachurch political activism because it became part of restorationist Neopentecostal theology due to a subtle, but fundamental, change in eschatology. By switching from a *premillennial* to a *postmillennial* posture, where the church must obtain a hegemonic control of society before Christ's second coming, a new age of activism was started in order to achieve dominion. It is important to clarify that most of the ideas that relate dominion theology to political theology were worked out by non-Pentecostal intellectuals within the so-called *Christian Restorationist* movement in the US (Garrard 2020), who formulated the main tenets in areas such as education, family law, economy, ecology, government, arts and entertainment, and others (McVicar 2015). However, Neopentecostal megachurch pastors and NAR leaders simplified the scope of their mission to what they called the *mandate of the seven mountains* (M7M) of society: *to go and gain hegemonic control of government, family, economy, education, religion, media, and arts/entertainment, in every nation of the world*, such that all areas of society submit to the rule of Christ upon his return (Rev. 11:15) (Enslow 2008).

Although this is a relatively new movement, many megachurches in Latin America adopted it as a missional statement. For example, the *Pacto de Caracas*, signed in 2010 by 70 apostles and megachurch pastors from all over Latin America, contains a contextualized version of the M7M (Wojtowicz et al. 2011). In this accord, ideas on how Latin American Evangelical megachurches and networks had to be involved in topics as diverse as politics, pluralism, the media, economy, health, family, creation care, social action, natural disasters, human rights, armed conflicts, justice, interracial society, arts,

and globalization, were established. For example, regarding regional political processes and their participation in them, the declaration contained in the final document clearly describes the dominionist agenda that animates Neopentecostal and NAR megachurches in the region, and the urgent need for these religious actors to get aggressively involved in politics (Wojtowicz et al. 2011, pp. 14–15). In the following 12 years, we saw the unfolding of a new Latin American Evangelicalism, led by megachurches, politically active and gaining power and public presence every day. As one researcher put it, "Evangelicals came to stay, stayed to grow, and grew to conquer" (Pérez Guadalupe 2017, p. 214). Megachurches and the NAR networks have been the main drivers of dominion theology in Latin America, influencing the vertiginous rise of Evangelical political activism in the last decade (Aguilar De La Cruz 2019). The change in the narrative of apostles, prophets, and megachurch pastors, committed to the mandate of the seven mountains (M7M), has served as a catalyst to participate in campaigns to elect Evangelical officials or sympathizers, as well as public protests about issues linked to the religious conservative agenda, such as opposition to abortion, same-sex marriage, and even some environmental issues.

By combining powerful leadership networks and a theology that imperatively justifies sociopolitical activism, and the attention of followers through advanced use of social media platforms (Córdoba Moreno 2021), a hyperactive involvement in politics is now commonplace among megachurches (Goldstein 2020). In Colombia, the political involvement of César and Claudia Castellanos is widely known (Pérez Guadalupe and Carranza 2021). In 1989 they founded their own political party, taking advantage of the religious capital that represented the growth of MCI due to its innovative G-12 cell model. As the G-12 model advanced, the capacity for political mobilization also grew, contributing to the Colombian right-wing agenda (Beltrán and Quiroga 2017). The influence of the megachurch was demonstrated in quite a palpable way during the plebiscite for the Colombian peace accords on October 2 (2016) (Bartel 2016a), where the influence of Evangelical megachurches was decisive for the success of the NO campaign (Beltrán and Creely 2018). O'Neill (2010, p. 62), describes the cell groups of the Guatemalan megachurch *El Shaddai* as a combination of "classroom, self-help group, and emotional laboratory" where a form of Christian citizenship is developed, a place where contemplation becomes action. Other authors have observed that the discipline that the cell groups promote helps megachurch pastors to mobilize believers to participate in national politics (Reu 2019; Algranti 2012). For example, in the *Alas de Águila* church in Altos de Chiapas (Mexico), through a transformation of the G-12 model, pastor Esdras Alonso created the *Army of God*, a kind of elite squad with military ranks, whose function is to actively participate in regional and national politics (Feria Arroyo 2017).

The transformation into votes of the religious capital of megachurches has been fundamental to promote a neoconservative Evangelical agenda in Latin America (Barrera Rivera 2021), with some electoral success and several setbacks as well. One of the countries where the implementation of dominion theology through the M7M has been progressively attempted is Guatemala. Starting with the government of Efraín Rios-Montt (between 1982–1983) and later during the presidency of Jorge Serrano Elías (between 1991–1993), the ideal of building a Christian nation has been present in Guatemalan politics. When Jimmy Morales, a self-declared Christian nationalist, was elected as president in 2016, megachurches lobbied to approve different family protection laws and the change of the Guatemalan embassy from Tel-Aviv to Jerusalem (Althoff 2021); and by alleging that its members were communists, anti-Zionists and promoters of gender ideology expelled the UN International Commission Against Impunity in Guatemala (CICG), which had been operating since 2006 as an autonomous body collaborating in complex cases of corruption and human rights violations (Aguilar De La Cruz 2019). More recently, the new Guatemalan President Alejandro Giammattei signed the approval by the congress of a *Public Policy Law for the Protection of Life and the Institutionality of the Family* with the support of the megachurches and Evangelical denominations. This law penalizes abortions with a minimum of up to five years in prison, apart from combating any group that is "inconsistent with Christian morality", curtailing

the rights of the LGBTQ+ population, declaring Guatemala the "pro-life capital of Ibero-America", and as a country "without the right to abortion" (Barrientos and Pérez 2022).

The implementation of the M7M by megachurches can also be extremely contradictory, and to some extent dangerous sometimes. Achieving a leading role in the political arena may imply the display of conflicting ideas, such as some megachurches supporting leftist governments, as happened with Hugo Chávez in Venezuela; or on the contrary, aligning with right-wing politicians such as in the case of former Peruvian presidential candidate Keiko Fujimori's support from Peruvian megachurches *Comunidad Cristiana Agua Viva* and *Movimiento Misionero Mundial* along with other pastors (Amat y León Pérez and Condor Vargas 2021). In the first case, in one of the few investigations on the politicization of Evangelicals in Venezuela during the government of Hugo Chávez, megachurch pastor Guido Raúl Avila, leader of the *REDIMA* international apostolic network, after the failed *coup d'état* in April 2002, stated his complete support for Chávez believing that "God is in control of Venezuela" (Smilde 2004). Avila added that God had told him that he would be Chávez' pastor, something that he tried, even organizing large prayer events in his favor, along with other megachurch pastors and apostles such as Jorge Porras Benedetti from Maracaibo and Jesús Pérez from *Catedral Renacer* in Caracas. Another contradictory position by a megachurch is the case of Colombian *Iglesia de Dios Ministerial de Jesucristo Internacional* (IDMJI), which through its political party MIRA was one of the few Evangelical churches that supported the SI to the peace agreements in 2016, but which later on, in 2018, backed up the election of right-wing candidate Iván Duque and decided not to support the peace bill in the congress, because of its gender-inclusive language (Coral Gómez 2020). IDMJI is one of the few Neopentecostal megachurches led by a woman in Latin America, but it holds views that oppose women's sexual and reproductive rights and LGBTQ+ rights.

## 5. Discussion

This article has reviewed a representative sample of the published research about megachurches in Latin America. One of the first observations is the clear imbalance in the study of megachurches in Spanish-speaking Latin America. Although there are similarities throughout the region, researchers need to recognize the varied paths taken by Evangelicals in each country to better understand how megachurches and their diverse practices have taken root in the different corners of the region. One aspect to consider is the uneven geographical distribution of the publications, with Argentina (31), Colombia (15), Guatemala (10), and Mexico (10) contributing to over 50% of the country-specific works. Other publications (28) were classified as considering topics covering regional issues in Central America or the South Cone. Guatemala was the first Latin American country where Evangelicals entered in the political arena occupying governmental offices, in part due to the role that megachurches played in the process. Argentina was fundamental for the Pentecostalization of Latin American churches, through its revivalist Evangelical culture that was appealing to the middle classes of a continent that was opening to new religious expressions. Colombia went from a very small percentage of Evangelicals to close to 20% in just a few years, due to the activism of many megachurches all over the country, which created methods of church growth that influenced congregations all over the world. In the case of Mexico, megachurches have been influential in worship innovation and new church paradigms, which have called the attention of researchers. Some of the journalistic research projects have expanded the scope to other regions and countries such as Peru, Venezuela, Chile, Honduras, and Costa Rica, but there is still a lack of academic research coming from these nations. This emphasis on a few countries has overshadowed the role of many other megachurches around the region, which also need to be studied in their context.

According to the research by *Leadership Network* (Bird n.d.), of the twenty largest Evangelical churches in Latin America (see Table 1), all except for one are Pentecostal/Neopentecostal, and surprisingly, one belongs to the new Postdenominational trend mentioned previously, but it also has a Neopentecostal background. Another important observation is that the majority were founded after 1980, which corresponds to the period of fastest and largest

growth of Evangelical congregations in the region. Certainly, there is a very strong link between Pentecostalism and church planting and growth that has served as a catalyst for this extraordinary expansion.

The theological and ecclesiological evolution of megachurches is another important aspect that needs to be investigated. Most of the articles reviewed classify megachurches within Neopentecostalism (Tec-López 2020); however, for many of the flagship congregations, there has been an evolution towards becoming *neoapostolic* churches, which implies quite drastic changes in theology, particularly the adoption of restorationism as a guiding principle. This means the weakening of denominationalism, or its total disappearance, and the formation of continuously expanding and interconnecting church networks, in many cases with megachurches serving as hubs. Christerson and Flory (2017) refer to this new movement as Independent Network Christianity (INC). Networks arise from large congregations or megachurches, around which other smaller churches that have been planted or adopted decide to associate under the leadership and coverage of the apostle or main leader of the umbrella congregation (Kay 2006). Also, *ritual exchange* can be seen as a facilitator to the formation of church networks (Robbins 2009), such as the networks formed in northern Mexico around Christian music as an element of ritual exchange (Ibarra 2019).

Another element that has been part of ritual exchange is the methodology of cell groups. Unfortunately, there is no mention of the G-12 model as a catalyst of networks in Christerson and Flory's study,[9] and in many other scholarly works about megachurches. Perhaps, as suggested by Garrard (2021, p. 206), this is because the G-12 model is the product of "South-South religious transnationalism". Currently, there is a critique on megachurch scholarship because it does not challenge the "self-sufficiency of the West and its cultural forms" (Mellquist Lehto 2021, p. 61). For that reason, despite many studies mentioning MCI, almost no attention has been devoted to its dense network, and how it has maintained its cohesion and progressive global expansion over time. In an earlier work, Alves showed the connections of a network of Latin American leaders instrumentalized from G-12 events, where the nodes of Castellanos, Anacondia, Cabrera Jr., Freidzon, and Huber, to name a few, stand out (Alves 2011). This study carried out more than a decade ago needs to be updated using the new technological and social network tools that have been developed, not only for the case of the G-12, but for other emerging Latin American megachurch and apostolic networks.

One unifying element has been the popularization of the new teachings of dominion theology. The COVID-19 pandemic proved to be fundamental for the strengthening of these networks around conspiracy theories, many of which use the dominion language of reconstructing society, leading the nations, and creation stewardship, according to biblical laws before Christ's second coming. The involvement in politics by Evangelicals, with the increasing sophistication of dominion theology, has been transformed into a new missional target for megachurches. It is no longer enough to carry out SM/SLSW; according to the new narrative, it is necessary to conquer the seven mountains of society, through any possible means. In the words of Edir Macedo, founder of the *IURD* in Brazil, it is a God-given "political nation-building project" (Pérez Guadalupe 2017, p. 11). To be fair, not all Latin American megachurches get involved in large-scale electoral politics, but they try to find space developing long-term sustainable social projects. However, it would be important to conduct research by Latin Americans to document these projects and their social impact considering the complexity of poverty and marginalization in these nations.

Progressively, some aspects of Pentecostalism present in the small *barrio* churches became more palatable for the open market of middle-class Latin Americans. Latin Americans from any social background were attracted to spiritual experiences, including wild visitations of the Holy Spirit manifested in being "slayed" by its power, incontrollable laughter, prophetic words, deliverance, and miraculous healings. For several years, the presence of these manifestations was considered a sign of God's blessing upon a church. But some form of routinization has been on the making over the years, bringing up questions that require answers. Have these experiences been repackaged by Latin American megachurches in a

different format? For example, the G-12 franchise includes a series of retreats called *Encuentros*, yet there is minimal research about these intense spiritual events, their specific impact, and how they relate with the overall culture of the megachurch. Have they changed over time? It is also important to find out if other programs are being implemented that come from other global megachurches that have routinized the charisma of their revival times (McClymond 2016). Are there special programs or spaces outside of church services where Pentecostal/Charismatic experiences can be experienced? As megachurch services in Latin America become more "friendly", like conferences with good music, where the spiritual needs of people can be satisfied in an amusing and less intense way, the "wilder" spiritual manifestations and gifts' expressions must be moved to more private environments. This seems to be the trend of some megachurches around the world, such as Hillsong, which is a Pentecostal church at its core, but where you rarely see this kind of display in any of their services (Klaver 2021). Many of the new PD congregations represent a mutation of the Neopentecostal churches that originated them, changing the liturgical structure and the soundscape of their gatherings (Ibarra 2019, p. 282). Lastly, have megachurches given more space to new healing methodologies that revolve around self-help, psychological inner healing, or coaching? This goes hand-in-hand with the proliferation of growth theologies where individuals are encouraged to seek individualistic solutions to their issues (Semán 2005), obtained from sermons, videos, and books of megachurch pastors who act as therapeutic experts in a wide variety of fields (Rocca 2013), especially in what has been called *inner healing*, which draws heavily from psychoanalysis and self-help therapies (Algranti 2008a). For example, in the case of *Más Vida* (México), there is a syncretism of ideas that combines "self-help, coaching, modern secular influences, globalized pop culture, supernatural beliefs, and biblical issues" (Gomes Rego 2020).

The lack of research on CWM is surprising, considering its widespread popularity. In one of the few studies, Gladwin (2015) showed how Latin American CWM contributed to make Neopentecostalism more attractive to middle classes and to younger generations of churchgoers. Mariela Mosqueira also makes a detailed account of how music and youth culture became fundamental for the spread of *La Unción* in Argentina, and the growth of many megachurches in the South Cone (Mosqueira 2016). Megachurches provided the space for worship leaders to compose and produce music, and to experiment with new styles and rhythms. Many worship bands have come from well-known Latin American megachurches, influencing all the region with their music, but very little attention has been paid to their contributions to Evangelical expansion. Even the enduring work of Marcos Witt, well-known pioneer of CWM in Latin America, has not been appropriately studied, except for a handful of works (Vélez-Caro and Mansilla 2020; Gladwin 2015). Due to its origin in the Davidic worship movement, Witt's theology advocates for "an anointed priestly and professional praise and worship" that "creates a charismatic classism and clericalism" (Gladwin 2015, p. 210) that permeates the megachurch movement, making CWM an instrument for the making of a certain type of Evangelical celebrity culture. Perhaps this is one of the reasons why megachurches' music lacks understanding of social struggles and of the perils of the marginalized or poor, and rarely, if ever, makes any critique to Latin American social inequalities. This becomes worse when megachurches adopt standardized worship music from *Hillsong*, *Bethel Music*, *Vineyard*, or *Elevation Worship*, because this translated music was originally composed in a whole different social context. In a similar way, very few attempts to understand the value and influence of translated and globalized CWM in Latin America have been published (Mora 2022c).

Cell groups as *social spaces* have received little attention in the literature from an organizational standpoint (Frigerio 2020). In a study in Argentina among young believers (Mosqueira 2019), 86.3% of respondents opted "for a type of pastoral care that is more sensitive and intimate, prioritizing face-to-face relationships and reciprocity" (p. 157). In addition, cell groups create social capital through the interconnections inside and across groups, which can help congregants in "navigating migration, securing jobs, negotiating politics, and so on" (Bauman 2022, p. 123). According to O'Neill (2010) the cell groups of *El*

*Shaddai* go beyond simple religious meetings, "allowing believers to craft a sense of self, to link that self to the fate of their nation" (p. 62). The discipline that cell groups promote, as well as the leadership that is exercised at the internal level of the group, added to the training that the network provides, are spaces where habits of thought and behavior are forged (Reu 2019), producing some form of *Christian citizenship* that transforms contemplation into action. Regarding the role of Evangelicals in Argentine politics, Algranti (2012) suggested that the most important impact of megachurches was at the level of the small groups, where cell leaders can encourage their disciples to participate in the political scene, beginning at the local or municipal level.

At a time when many megachurch pastors are considering starting a political career, the scarce number of papers addressing their leadership from a more theoretical standpoint is surprising. We can see that megachurches can be very complex organizations, where leadership practices are fundamental. For the most part, most articles classify megachurch leadership simply as *charismatic* in the Weberian sense, but also in the *pneumatological* sense, as these pastors and apostles must also demonstrate *signs and wonders* in their ministry. This means that in the demonstration of leadership skills, there is an *organizational side* and a performative or *embodied side* that is public and visible. The term *pastorpreneurs* serves to describe this emerging ecclesial leadership class, proposing an "aesthetic approach" to study "the ways pastors authenticate their leadership through their bodily performances, observed on stage and screens, during church services and in the online world" (Klaver 2015). The story or testimony of the megachurch pastors, told in dramatic ways, following the steps of biblical prophets, is an important source of leadership authenticity and divine calling, and the basis for the vision of the megachurch. Perhaps there is a commonality in Latin American pastors when they describe their transit through the different stages of *crisis, awakening, mission, objection*, and *signals of confirmation* that characterize their vocational calling to become pastors of multitudes, since most of them started their ministries from the margins, trying to avoid denominational control, and competing with the hegemony of the Catholic church. The study of the rich variety of these *crucible experiences* in the formation of megachurch pastors as leaders has not been attempted yet (Bennis and Thomas 2002). Washington et al. (2014) have suggested that through the practice of book writing, megachurch pastors exert "control over storytelling", which determines the vision and mission of the church. Since some Latin American megachurch pastors have been very prolific in book writing (although just a few have had a worldwide impact), the cultural offer of the megachurches bears the mark of the pastor or apostle, his family, and the church story, in a variety of products "that express conformity with the style and purpose offered by the institution" (Algranti 2015). The study of megachurch leadership authentication through books, video streaming, conferences, and networking is also an open area of research, in terms of understanding their views of leadership, theology, ecclesiology, organizational dynamics, and politics through their written stories, and reflections, as well as embodied messages.

Latin American cultural models of leadership such as authoritarianism, paternalism, machismo, power hunger, risk avoidance, and uncertainty control (Romero 2004) require serious consideration among megachurch leadership. In a continent that has produced the likes of Fidel Castro, Augusto Pinochet, Hugo Chávez, Alvaro Uribe, Lula Da Silva, Cristina Fernández, Nicolás Maduro, Jair Bolsonaro, or Andrés López Obrador, it would be interesting to find out how different are megachurch pastors from these regional leadership stereotypes. For the most part, the literature tends to portray the Latin American leader as an "authoritarian-benevolent paternalistic figure that leads in a parental way, that engenders care and loyalty", and follower submission (Castaño et al. 2015). For Ronny Chaves, the leadership of megachurch pastors provides some sort of *apostolic fatherhood covering* to the church, since they are "fathers" who bring healing and are primarily interested in the wellbeing of their "spiritual children". For him, this is one of the strategies of the Holy Spirit to heal the *father wound* of potential leaders, as a first step to building a better nation through Christian dominion (Chaves 2004, p. 20). However, given the size of the

churches, this new father figure seems still distant; only to be admired, revered, and obeyed through sermons, videos, books, tweets, and by following the vision of conquering the seven mountains of society.

Less researched still are the characteristics of female megachurch pastors and leaders, many of them with stories of unending struggle in a field dominated by patriarchy. In the list of the 50 largest Latin American churches there are no women pastors; when women are mentioned in this list, it is in relation to the married couple. This is a widespread phenomenon; for example, in the USA, less than 1% of American megachurches are led by a female senior pastor (Mathews 2022). Referring to Chilean Neopentecostalism, Mansilla (2007) classifies female pastoral ministry in three phases: *penumbral*, legitimized by the position of the husband as senior pastor or apostle; *supplementary*, where women accidentally access the leadership position due to the lack of men in ministerial functions; and *autarkic*, in which the woman reaches her position on her own account. Whatever the phase considered, women's access to pastoral and leadership positions occurs through the most varied and unthinkable paths, disrupting patriarchal organizational dynamics, and constructing their leadership identity within a power struggle that crafts their subjectivity over a lifelong itinerary (Méndez and Mora 2013).

The penumbral phase is the most common access to the female pastorate, as many prominent women pastors are married to men who occupy high hierarchical positions, are founders of megachurches, or lead apostolic networks (Machado 2005). Usually, they act as *spiritual mothers*, providing maternal care to all areas of the church where they participate (Méndez 2013), reaffirming her authority with the title of *Pastora* (Algranti 2007). Even in the penumbral phase, women must discover ways to become accomplished ministers in the religious sphere, and submissive wives in the privacy of home, in what has been termed as *paradoxical domesticity* (Madrazo 2017). These are women that constantly negotiate their preaching acuity, political abilities, entrepreneurial capabilities, and their social sensitivities with submission to the men of the church (Bowler 2019). In many cases, they are recognized as leaders but not as megachurch pastors; they take titles such as church administrators, prophets, teachers, advisors, managers, coaches, politicians, senators, and entrepreneurs, so long as they do not trespass the authority of the husband and keep family as their priority (Ramos 2021, p. 204).

The recent designation of women pastors in two of the largest seeker-sensitive megachurches in the USA (*Saddleback Church* and *Willow Creek Community Church*) will make the topic of women leadership in megachurches an important area of research. A recent study suggests that PD churches in Mexico are open to female leadership (Ibarra and Gomes 2021) but mentions only a *treasurer/accountant* and a *chief designer* as examples of female leadership. It is likely that the Neopentecostal background of some pastors of PD churches weighs heavily when deciding to innovate and promote women to senior leadership positions. This is the reason why many women leaders, frustrated by the male-dominated religious hierarchy, decide to live their "faith" in alternative ways, breaking with their original congregations and founding new churches or networks. These autarkic women pastors and their churches need to be investigated in comparison with those led by men. Along the same lines, many women develop ministries without the impositions of patriarchal ecclesiastical organizations, such as independent digital ministries, using social networks and Internet tools, which attract a sizable number of followers and with values that are compatible with megachurch theology and practices (Gaddini 2021).

Also, due to the large diaspora and migration dynamics that exist throughout Latin America, and from the region to the USA and Europe, it is important to study the new megachurches established by this moving population, considering that many Latin American megachurches have started branches mainly in Florida, California, New York, Texas, Spain, Italy, Portugal, and the UK (Oro 2014). Better known in the literature as *reverse mission* (Burgess 2020), it refers to the efforts of churches in the Global South, which were once subject to missionary enterprises, to become senders of church planters to secularized countries in the Global North (Oro 2019). There are several themes to explore in

this line of research regarding megachurches, such as the theological reasons for such efforts (expressed rhetorically as "Christianizing Europe again"), the economic implications involved in terms of financial resources for reverse mission projects, and the importance for these sending churches of becoming an influential force with international recognition. While most of the research about reverse mission has centered on Nigerian, Korean, and Brazilian megachurches (Burgess 2020), there are several other Latin American examples, particularly developed through neoapostolic networks such as G-12, that would be worth researching in the future. Besides the more formal church-planting projects undertaken by megachurches, one interesting direction to explore is the growing dynamic flow of relations and interactions that diasporic communities have with churches in their mother land, using the digital space. These new communication channels allow for constant communication between megachurches and their migrant members, who may end up starting cell groups or new churches wherever they go with support from their mother churches. In this way, the reach of megachurches goes beyond physical spaces to virtual dimensions that change the logic of worship, preaching, and pastoring as had been practiced. An example of this new expression of *digital religion* among diasporic communities has been recently explored as a case study of Venezuelan Evangelical migrants during the pandemics (Mora and García 2022), but there is ample room for more ethnographic work in this area.

The growing problem of misconduct, corruption and scandals provoked by Latin American megachurches is the last topic to discuss. Even though many charismatic megachurch pastors, apostles and prophets can start, lead, and reproduce successful organizations and movements, the tendency toward failure is higher than in other types of leadership models (Oakes 2010). One of the few works that not only documents the amazing megachurch growth in Brazil but also considers the criticisms and shortcomings of these huge ministries is the article by Smith and Campos (2015). Nevertheless, in recent years, the most common critical assessment has been made through journalistic investigations about the wrongdoings of several Latin American megachurch leaders (Ward et al. 2020), and their contradictory, sometimes corrupt, involvement in national politics (Torres et al. 2019). Also, popular-culture portrayals of corrupt Latin American megachurch pastors are growing in the region, either in novels (Gamboa 2019), or in popular streaming video series (Galindo 2021). Issues of sexual misconduct, money laundering, abuse, and vote selling, are often mentioned in Latin America, but there is still an absence of serious academic studies on the subject. The looseness of megachurch network affiliation exacerbates the lack of accountability, enables the concentration of power in few individuals, limits megachurch organization to function mainly as family ventures with nepotistic leadership transitions, and hides the emergence of narcissistic personality disorder in the leadership, all of which could eventually lead to church problems, including criminal activities.

## 6. Conclusions

This review has demonstrated that the field of megachurch studies in Latin America is an open field of research. For the most part, the scholarship available has come from the sociological and anthropological studies of Neopentecostalism by research groups in universities in the region, therefore lacking strong ecclesiological and theological argumentations. It is very important to view megachurch studies from a regional and global perspective, overcoming the thought that Latin American megachurches are mere clones of North American prototypes. This article has shown that there are many examples that make the field unique, by considering original church growth methodologies developed in the region, the way Pentecostalization of Latin American Protestantism has occurred, and how theologies of power have been adapted to Latin American idiosyncrasies. In this regard, the theological basis underlying the priorities and practices of any megachurch must be considered more carefully, making it necessary in future studies to modify the definition of a megachurch from the purely numerical (>2000) to one that also incorporates how the church understands and engages with society, and what initiatives it takes to change it. There is a vast room for ethnographical theological research to understand the

many changes that have made Evangelicals in Latin America evolve, from the thought of the church as a "refuge", to a new political activism that seeks to bring Christian dominion of countries with corrupt and unjust governments, which would allow the reign of God to be established on Earth, paving the way for the second coming of Jesus.

**Funding:** This research received no external funding.

**Institutional Review Board Statement:** Not Applicable.

**Informed Consent Statement:** Not Applicable.

**Data Availability Statement:** Not Applicable.

**Acknowledgments:** The author wishes to express gratitude to the peer-reviewers and the special issue academic editor for their valuable comments that improved greatly the content of the article. *Muchas Gracias*.

**Conflicts of Interest:** The author declares no conflict of interest.

## Notes

1. In some Latin American countries, civil wars left some towns without priests for many years, giving rise to a form of popular religion, a syncretism of Catholicism, African, and Amerindian beliefs.
2. See for instance the 2020 Vol. 3 (6) of *Encartes Journal* from Mexico, and the 2021 Vol. 23 of *Ciencias Sociales y Religión* from Brazil.
3. About the hermeneutics of the FTL see for instance (Ávila Arteaga 2011).
4. It is important to note that the evolution of Pentecostalism in Brazil, as well as in many other Latin American countries such as Argentina and Chile, is seen as a progression of three waves, namely a *first* wave of foreign missions; a *second* wave of quadrangular faith (Jesus is seen as a savior, sanctifier, healer, and who empowers the believer) that facilitated the birth of indigenous churches and denominations; and a *third* wave of urban mission, social presence, and where the gospel of prosperity is added to the previous four dimensions (Freston 2019). The third wave was the driver for the expansion of megachurches.
5. Wariboko and Oliverio (2020), indicate that there are 644 million Pentecostals/Charismatics (P/C) worldwide, which means that one in twelve persons today is a Pentecostal or Charismatic Christian. There are 195 million Pentecostal/Charismatics in Latin America alone. This would give a 29% of Latin Americans considered as P/C, a figure 6% higher than the 2006 PRC study.
6. In 1997, acting on prophetic words, Chaves and Ana Méndez from México organized an expedition to the base camp of Mount Everest, some 5400 m high (although the original goal was to reach the top), to carry out what they called *Operation Ice Castle*. By performing SLSW with a team of 26 intercessors, they expected to defeat the demonic fortress of the *Queen of Heaven*, which according to their SM was located territorially in the Himalayas.
7. In the mid-1980s, a Christian worship music scene emerged in Mexico, made up of several names, groups, songs, and albums from that country. Among the initiators are Marcos Witt, along with Marcos Barrientos, Jesús Adrián Romero, Cesar Garza, Coalo Zamorano, Lorena Warren, Edgar Rocha, Emmanuel Espinoza, Miguel Cassina, the *Torre Fuerte* group made up of Héctor Hermosillo, Heriberto Hermosillo, and Álvaro López, as well as other Central American worship leaders such as Danilo Montero, Juan Carlos Alvarado, and Jaime Murrell, who somehow shared ideas, resources, and productions for several years, in what could be considered as the incipient Latin American worship and praise scene. Independent record labels also appeared to promote such music, such as Marcos Witt's *CanZion Producciones*, Marcos Barrientos' *Aliento Music Group*, and Jesús Adrián Romero's *Vástago Producciones*, as well as some well-known evangelical publishing houses that had an interest in P&W music, such as *Editorial Vida*.
8. A brief analysis of the writings of Marcos Witt shows that they emphasize, among other things, the typological model of the tabernacle of David, relying on in Psalm 22:3 and Psalm 100; they highlight the preponderant role of musicians as *Levites* within the organization of the music ministry of the church; they motivate the search for the presence of God through P&W music; they describe the spiritual warfare that takes place through music; apart from touching on other topics such as spontaneous worship and prophetic songs (Witt 1993, pp. 203–18).
9. They limited their work to those networks that originated in the United States, beginning with Bethel Church and Harvest Church in California and IHOP in Kansas City.

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
