# Peer review of "Latin American Megachurches in a Changing Culture: An Integrative Review and an Exploration of Future Research Directions"

_religions, doi:10.3390/rel13090843_

Round 1

Reviewer 1 Report

The author(s) raises interesting and worthwhile questions about scholarly gaps in research on megachurches in Latin America. The focus on issues of organization and social interaction is most welcome. As the authors rightly note, very few studies address these topics. I would also point out that the article’s thorough review of the secondary literature is impressive, particularly its references to Spanish and Portuguese language sources. These studies are seriously underutilized. However, the paper has several minor issues (below) that should be resolved prior to publication.

To begin, the article needs a more nuanced discussion of the term ‘evangelical’ as it is used in Latin America. For scholars unfamiliar with the historical development of the movement in the region, it would be useful to include (even in a footnote) a more complete description of why evangélico has become a colloquial, catch-all term that includes most (but not all) non-Catholic Christian groups. I suggest using Kenneth Pike’s terminology (emic), to describe it as a local description of non-Catholic Christian believers.

The overview of the work of Lalive d’Epinay and Emilio Willems on page 2 needs additional discussion. While their work has been extremely influential in the field, it has not gone unchallenged. Their overreliance on ‘anomie’ as a framework for understanding the growth of Pentecostalism has attracted great criticism in recent work. The author(s) might consider referencing many of the critiques that emerged following Jeffery Rubin et. al.’s special issue on lived religion in the Latin American Research Review (2014).

Likewise, I suggest the author(s) consider integrating more recent data from the Pew Research Center’s report on Religion in Latin America (2014) rather than 2006.

One area for significant improvement throughout the paper can be found in the author’s broad assertion that research on megachurches in Latin America is geographically unbalanced (focusing mainly on Argentina, Colombia, and Central America) because these countries “brought some innovation” (p. 19, ln 893). Such a claim overlooks the diverse historical development of evangelicalism throughout Latin America. One clear outlier to this is Chile, where the ‘pride’ of local independence from foreign missionary work dominated the movement from the beginning. More recognition of the varied paths taken by evangelicals within each country is needed to better understand why megachurches and their diverse practices have not taken root in all corners of the region.

Also, the reference to the Jotabeche Cathedral in Chile in Table 1 should be changed to Primera Iglesia Metodista Pentecostal de Chile. Jotabeche is the name of the cathedral, but not the name of the church.

Finally, the article needs minor stylistic and grammatical revision.

Author Response

Reviewer 1

  1. As suggested, I have added a better definition of the term ‘evangelical’ as it is used in Latin America. I have added a reference to support the claim that Pentecostals inherited the main theological characteristics of Evangelicalism.
  2. Regarding the work of Lalive d’Epinay and Emilio Willems I have expanded the discussion. Besides Rubin et. al.’s reference, I have also included others and have framed better this aspect of the research of Pentecostalism in the region.
  3. I had cited the 2014 PRC study, but I included it when I mention the frowth of Pentecostalism in the region. Besides that, I added some figures from the World Christian Encyclopedia (3rd Edition).
  4. I have made clearer the varied ways that Evangelical churches have evolved in the region. Also, I clarified the statement about the research on megachurches in Latin America being geographically unbalanced.
  5. I changed Table 1 using Primera Iglesia Metodista Pentecostal of Santiago de Chile for the megachurch.

Reviewer 2 Report

This is an excellent article on a very important topic: megachurches in Latin America.  Its list of References is amazing in its length and commendable for its ample and systematic covering of the key literature in Spanish and Portuguese.  The author is absolutely correct that essential literature in these languages is often overlooked in the literature originating in Europe and the USA.  The author’s systematic review of 131 references also brings out the uneven geographical distribution of the publications, with Argentina (24%), Colombia (11.5%), Guatemala (7.5%), and Mexico (7.5%) making up over 50% of the total (p. 5).  (I’m surprised Brazil is missing here, perhaps as a result of the decision to leave out the Universal Church of the Kingdom of God.)

I agree with the selection of the “three broad catalysts to the emergence of megachurches in Latin America: Church growth methods, Pentecostalization, and Theologies of growth” (p. 5).  However, there is overlap between the first and the last.  New methods and relevant data on megachurches might be distinguished easier by perhaps following the 10 main topics from Bauman (2022): geography, growth, worship, leadership, demographics, gender and sexuality, etc. (see p. 4).  Most of these topics now appear in a more random and less structured way.

Catalyst 1 (Church growth methods, pp. 5-9) is described chronologically; for catalysts 2 (Pentecostalization, pp. 9-14) and 3 (Theologies of growth, pp. 15-19) the chronology of the sequence is less clear.

There are a few sweeping statements and generalizations without a clear basis in evidence and references.  For example, on p. 12 in line 564-565: “at some point in their history, every megachurch in the region engaged in SM/SLSW” [spiritual mapping/strategic level spiritual warfare].  Please provide data and references to back up this statement.  It’s hard to believe that this is true for every megachurch in Latin America.  From personal observation, I know it’s not true for the Centro Familiar de Adoración megachurch in Asunción, Paraguay (listed in Table 1, p. 10).  However, there is certainly spiritual warfare going on in some of its 1,500 cell groups.  Perhaps the easy fix would be the change the original sentence into: “at some point in their history, cell groups in every megachurch in the region engaged in SM/SLSW”.”  Even better would be to provide some data and sources to back up this sweeping statement.

The description of Dominion Theology (p. 17, line 784) lacks precision and clarity; perhaps a succinct definition could be added?  I don’t believe that the election of Jimmy Morales as president in Guatemala is due to a “successful implementation” of Dominion Theology (p. 18, line 851).  Please provide data and references to back up this statement.  Althoff (2021) lists several other important factors contributing to his election, including quite a few that are nonreligious.

Missing literature in the References:

p. 1, line 21: Cartledge et al. 2019

Typos and errors etc. that need to be corrected:

p. 2, line 57: Emilio Willems

p. 4, line 170: Undergraduate theses [plural]

p. 4, footnote 4: journal name Ciencias Sociales y Religión should be in italics

p. 5, line 239: Donald McGavran

p. 5, line 241: Donald McGavran

p. 10, line 474: charismata of the Spirit

p. 12, line 548: Garrard 2021

p. 12, line 567: Garrard 2021

p. 15, line 679: of blessing and a [missing word!] of sowing and reaping

p. 15, line 689: Garrard 2021

p. 19, line 925: many studies mentioning MCI

p. 20, line 940: conspiracy theories

p. 20, line 947: it is a God-given

p. 20, line 962: called Encuentros, yet there is [or: however,]

p. 21, line 986: The lack of research on CWM is surprising, considering its widespread popularity.

p. 21, line 1016-1017: wrong font size

p. 21, line 1037: dramatics ways

p. 27, line 1360: Donald McGavran

Author Response

Reviewer 2

  1. I have deliberately left Brazilian megachurches pretty much out of this review; I think that the topic demands a full paper for that alone. This has been clarified accordingly.
  2. I have explained in the Methodology section the reasons to concentrate in three broad catalysts to the emergence of megachurches in Latin America instead of using the 10 topics by Bauman. Basically, because there are not enough research efforts in all the categories to make them meaningful. I agree that there could be overlap between the first and the last catalyst, however, they appeared in different times and Church Growth was embraced before Pentecostalization was prevalent in the region. Also, I tried to emphasize the theological aspect and for that reason proposed the last catalyst in this direction.
  3. I agree that for Catalyst 2 (Pentecostalization) the chronology of the sequence is less clear. This is because the number of topics is larger and there are some of them such as CWM that have not been studied sufficiently, there is not even a history of worship music in the region, despite the overwhelming diffusion of it throughout the continent. However, for Catalyst 3, I tried to present the theological evolution which started with Prosperity Theology, then it was amplified with the idea of Wealth Transfer, but the latest development is the popularization of Dominion Theology.
  4. I have tried to minimize the generalizations as much as possible.  I changed one of the texts to the suggested phrase. There is not much data regarding Spiritual Warfare practices in Latin America, I think this would be an area for further study.
  5. I expanded the description of Dominion Theology an added another reference to better support it.  
  6. Perhaps the way the paragraph was written implied that the election of Jimmy Morales as president in Guatemala was due to a “successful implementation” of Dominion Theology, this was not my intention and has been modified accordingly. I think the idea was to show how dominionism has been a growing force in Guatemalan politics.
  7. All other corrections suggested were taken into consideration.

Reviewer 3 Report

Excellent and relevant article. It clearly highlights the importance of this topic and opens the door for future research. I suggest to include a list of potential topics for further research. Also, more charts (similar to the one included) summarizing key content and churches would help the reader.

Author Response

Reviewer 3

  1. I suggested in the conclusion some general areas of potential further research. But I think a full detailed list would be more adequate for a workshop or a working meeting of interested parties, for which this article could be quite useful.
  2. Regarding the second suggestion, I would prefer not to change the style of the article at this point.